# Advances in the Modification of Silane-Based Sol-Gel Coating to Improve the Corrosion Resistance of Magnesium Alloys

**DOI:** 10.3390/molecules28062563

**Published:** 2023-03-11

**Authors:** Jiao Li, Huanhuan Bai, Zhiyuan Feng

**Affiliations:** 1School of Chemical Engineering and Technology, Sun Yat-sen University, Zhuhai 519082, China; lijiao35@mail2.sysu.edu.cn; 2Department of Chemical and Materials Engineering, University of Kentucky, Lexington, KY 40506, USA; huanhuan.bai@uky.edu

**Keywords:** sol-gel, corrosion, nanomaterials, Mg alloys, coating

## Abstract

As the lightest structural materials, magnesium (Mg) alloys play a significant role in vehicle weight reduction, aerospace, military equipment, energy saving, and emission reduction. However, the poor corrosion resistance of Mg alloys has become a bottleneck restricting its wide application. Developing a good surface protective coating can effectively improve the corrosion resistance of Mg alloys. The silane-based sol-gel coating technology has been widely used in the corrosion protection of Mg alloys in recent years due to its advantages of simple process, accessible tailoring of film composition and structure, and excellent corrosion resistance. Whereas the synthesis of sol-gel coatings includes the hydrolysis and dehydration process, which may inherently contain micron or nano defects in the coatings, thereby making it detrimental to the anti-corrosion effect. Therefore, in order to enhance their protection against corrosion, the appropriate modification of sol-gel coatings has become a current research hotspot. This review is based on the modification methods of silane-based sol-gels on the surface of Mg alloys, which are divided into four categories: bare sol-gel, nanoparticles, corrosion inhibitors, and sol-gel-based composite coatings. The modification methods and corrosion protection mechanism are discussed respectively, and the application, development, and research strategies of silane-based sol-gel coatings are included.

## 1. Introduction

Mg alloys are known as the lightest structural materials [1,2]. Compared with other metals, Mg and its alloys have the advantages of low density (lightweight), excellent biocompatibility, being environmentally friendly, good recyclability, having high specific strength, and having a specific stiffness and good damping performance [3,4,5]. Due to the above advantages, Mg alloys have broad application prospects in the fields of aerospace, the automobile industry, medical materials, national defense, and military equipment and communications [6,7,8,9].

The standard electrode potential of Mg is very low, which is −2.37 V_SHE_, and it has extremely high chemical and electrochemical activity [10,11]. Therefore, in contact with other metal materials, Mg alloys often act as anodes and are subjected to electrochemical corrosion [12,13]. Additionally, the Pilling–Bedworth (P–B) ratio ($\frac{Molecular\;volume\;of\;oxide}{Molecular\;volume\;of\;metal}$) of MgO is only about 0.81 < 1, which results in the formation of a porous and unprotective oxide film on Mg substrate and easily leads to pitting corrosion, stress corrosion cracking, galvanic corrosion, and other forms of corrosion [13,14]. Therefore, poor corrosion resistance is the main disadvantage that hinders the application of Mg and its alloys [13,15,16].

There are several approaches to increase the corrosion resistance of Mg and its alloys, including internal methods such as the modification of alloy composition/microstructure (fabrication of high-purity alloys, refinement of microstructure, etc.) [12,17,18,19,20]. The external methods include surface treatment and coating processes (electroplating, chemical conversion coating, electrochemical conversion coating, laser/ion or electron beam treatment, sol-gel coating, and organic coating, etc.) [6,21,22,23,24,25,26,27,28,29]. Among them, the surface coating method may be the most simple and direct method. The sol-gel coating is widely used to protect Mg alloys against corrosion due to the following advantages:No complicated equipment is needed.It can be synthesized at room temperature without vacuum conditions.Large-area thin films can be prepared on the surface of Mg.Modification of the coating composition and microstructure is simple. It is easy to prepare a uniform multi-component composite coating, and the thickness can be adjusted at the micron level.It can be used alone, as a pretreatment layer or as a sealing layer [30,31,32,33,34].

Typically, the sol-gel coatings are synthesized by using polymer gel methods. Sols are obtained through the hydrolysis and condensation reactions of compounds containing highly chemically active precursors, organic solutions (such as ethanol), and catalysts (such as weak acids and bases). The sol is aged for a period of time until the colloidal particles are aggregated, resulting in reduced fluidity. When the sol turns into a gel, the linear structure becomes a network structure, thus forming a polymer network. Then, through curing methods such as drying and heat treatment, it will eventually be deposited on the surface of the alloy [34,35,36]. In the synthesis of silane-based sol-gel coating, the most commonly adopted precursors are alkoxysilanes, such as tetraethoxysilane (TEOS) due to its relatively high thermal stability and moderate reactivity over a wide range of temperatures [37,38]. These precursors help the production of silane-based sol-gel achieve high-wear resistance, good adhesion, and corrosion protection [37]. Additionally, sol-gel technology can be used to prepare thin films, powder materials, bulk air-condensing materials, etc., and is widely used in the fields of anti-corrosion, biology, electricity, and optics [39,40,41].

Generally, the film formation of the silane-based sol-gel film on Mg alloys needs to go through three stages (hydrolysis, adsorption, and condensation) [42]. The schematic diagram of silane coating formation process is shown in Figure 1.

The silane-based sol-gel is rich in a cross-linked outer silane layer (Si–O–Si bonds), that can form a stable Mg–O–Si layer at the metal oxide/sol-gel interface providing a corrosion protection effect. [43,44]. Figure 2 is a schematic diagram of the steps and processes commonly used to obtain sol-gel coatings on Mg alloys.

When the silane-based sol-gel coatings are exposed to saline environments, although they have a good protective effect in the short term, they cannot provide sufficient anti-corrosion effects in a long run. The reason is that the synthesis of sol-gel coatings includes the hydrolysis and dehydration process, which may inherently contain micron or nano defects in the coatings, therefore making them detrimental to the anti-corrosion effect. They will eventually allow the electrolyte to diffuse to the coating/substrate interface, resulting in the corrosion of the substrate [39,45,46,47]. The corrosion protection of sol-gel coatings can be improved by increasing their thickness. However, if the sol-gel coating is too thick, cracking and delamination will occur due to the high residual stress on the surface during the drying and curing stages. Additionally, it reduces the adhesion between the sol-gel coating and the metal surface [48]. Several efforts have been made in recent years to improve the corrosion protection properties of sol-gel coatings. The incorporation of corrosion inhibitors into sol-gel coatings has emerged as a promising approach to overcome the above disadvantages [49]; using nanoparticles as a filler is also an effective way to increase corrosion protection [50,51]. Another advantageous way to achieve this goal is to use composite coatings, such as sol-gel as a pretreatment layer or as a sealing layer. Details of all these methods are within the scope of this review.

So far, there have been many excellent reviews on related topics such as the history and applications of the sol-gel process. However, these reviews mainly focus on the development history of sol-gel [30,35,45,47,52] and the basic principles [30,35,38,39,45,53,54], including self-healing, improved biocompatibility, antifouling, superhydrophobicity, etc. [30,35,38,45,52,53,54]. However, no attempts have been made with the perspective of silane-based sol-gel coating modification methods on Mg alloys. Herein, the focus of this article is on the modification methods, rather than the chemical synthesis process of sol-gel. The modification methods are divided into four categories: (1) Bare sol-gel; (2) Nanoparticles; (3) Corrosion inhibitors; and (4) Sol-gel-based composite coatings. This review provides a comprehensive account of the topic, while also offering some perspectives on future developments.

## 2. Bare Sol-Gel Coatings

Organic-inorganic hybrid (OIH) coatings prepared by the sol-gel method are suitable for corrosion protection [52,55]. OIH coatings combine the advantages of organic polymers (i.e., impact resistance, flexibility, and lightweight) with the properties of inorganic materials (i.e., high adhesion, chemical resistance, thermal stability, and mechanical strength) [35,52]. For example, Hu et al. studied the preparation of SiO_2_(TV) sol with TEOS and triethoxyvinylsilane (VTEO) as the precursors and compared its corrosion resistance with another SiO_2_ (T) sol prepared with only TEOS as a precursor on AZ91 Mg alloy [56]. The experimental results showed that the corrosion resistance of SiO_2_ (TV) sol coating was better than that of SiO_2_ (T) coating under immersion in aqueous 3.5 wt.% NaCl. Khramov et al. studied the synthesis of stable hybrid coatings with phosphonate functional groups through the sol-gel route of co-condensation of TEOS and diethylphosphonatoethyl-triethoxysilane (PHS) [57]. The improved corrosion protection of phosphonate-containing coatings, as compared to pure sol-gel coatings, had been observed and explained by the favorable combination of barrier properties of the organo–silicate matrix with strong chemical bonding of phosphonate groups to the Mg substrate. At the same time, during the preparation process, catalysts, solvents, aging time, processing temperature, etc. all have a great influence on the sol-gel coating. Hernández–Barrios et al. studied the influence of catalyst (acetic acid) concentration, immersion time, and aging time on the synthesis and deposition process of TEOS–GPTMS composite sol-gel coating on AZ31 Mg alloy, and analyzed the effect of these synthesis parameters on corrosion performance [58]. The experimental results showed that the gel kinetics produced by the catalyst concentration of 10 vol% was more stable. However, the low pH value promoted the formation of corrosion products and hydrogen on AZ31 alloy, which affected the morphological characteristics of the coating in the deposition stage. On the contrary, when the catalyst concentration was less than 5 vol% and the immersion time was equal to or less than 30 s, the obtained coating was continuous and uniform. Almost no corrosion products were observed at the substrate interface. Simultaneously, the chemical bonding between the Mg surface and the sol-gel network was achieved. On the other hand, the aging time was another factor affecting the quality of sol-gel. The aging time of 3 days and 6 days can get the optimum viscosity, pH value, and sol-gel reaction. It can fully densify the SiO_2_ network, reducing the formation of corrosion products and hydrogen. In conclusion, continuous, uniform, and dense sol-gel coating can be obtained by proper controlling of the synthesis parameters, which can slow down the surface corrosion process of AZ31 Mg alloy during sol deposition. Furthermore, corrosion tests showed that the corrosion current density was about an order of magnitude lower compared to AZ31 substrates in aqueous 0.1 M NaCl.

The above simple OIH coating can improve the adhesion between the substrate and the sol-gel coating, but it can only function as a physical barrier. Once the film layer is damaged, it will no longer have a protective effect until the film layer is completely separated from the substrate. As shown in Figure 3, the schematic diagram of the anti-corrosion principle of bare sol-gel coatings is presented. In order to obtain OIH sol-gel coating with better anti-corrosion performance, other additives must be introduced in the OIH network such as corrosion inhibitors and nanoparticles.

### 2.1. Bare Sol-Gel with Corrosion Inhibitor

The corrosion inhibitor-loaded sol-gel coating is mainly achieved by incorporating appropriate inhibitors in the sol preparation process [59,60]. These corrosion inhibitors are stored in the prepared sol-gel coating. The stored inhibitors can be released by the change in the coating status or the external environment, providing corrosion inhibition and a self-repairing effect on Mg alloys to further improve the corrosion resistance [61]. Currently, corrosion inhibitions mainly include inorganic salts and nitrogen-containing organic compounds. Cerium salt is one of the most commonly used inorganic corrosion inhibitors [62]. Zhong et al. studied the effect of cerium concentration on the microstructure, morphology, and anti-corrosion properties of the cerium-based sol-gel coating on AZ91D Mg alloy [63]. Five different concentrations of cerium nitrate hexahydrate were added as dopants to the sol-gel. It was determined that the degree of decomposition of silane chains in the sol-gel network improved with increasing cerium concentration. Cerium present in the sol-gel coatings exhibited the ability to deposit on the active sites, leading to the formation of corrosion products with greater protective features. This phenomenon can slow down the corrosion process and provide an additional protective effect on the substrate [1]. EIS results showed that the corrosion resistance of the coating increased first and then decreased with the increase in cerium concentration. When the cerium concentration was 0.01 M, corrosion resistance reached its maximum (R_p_ = 143 kΩ·cm^2^) in aqueous 3.5 wt.% NaCl. Afterward, Murillo–Gutiérrez et al. and Hernández–Barrios et al. performed similar research and reached the same conclusion [37,64].

Some organic compounds with special functional groups and structures are promising for corrosion inhibition. They mainly contain nitrogen, oxygen, sulfur, phosphorus, or aromatic rings in their structure. Galio et al. synthesized a composite sol-gel coating doped with 8-hydroxyquinoline (8-HQ) and applied it to Mg alloy AZ31 [61]. The positive effect of 8-HQ on corrosion protection can be explained by the formation of insoluble and stable complexes Mg(8-HQ)_2_ that block the propagation of corrosion and fill the microporous and micro defects in the sol-gel coating.

Amino acid is an environmentally friendly inhibitor that is relatively cheap and easy to prepare [65,66]. The corrosion inhibition ability of amino acids can be attributed to their propensity to form hydrogen bonds with oxides or hydroxides on metal surfaces [66]. Li et al. prepared a sol-gel coating containing the corrosion inhibitor paeonol condensation tyrosine (PCTyr) Schiff base on the surface of ZE21B Mg alloy to improve corrosion resistance and biocompatibility [67]. The i_corr_ value of PCTyr Schiff base sol-gel (3.64 × 10^−6^ A/cm^2^) was about two orders of magnitude lower than that of the substrate (1.31 × 10^−4^ A/cm^2^) in simulated body fluid (SBF). These results demonstrate the potential of Schiff base-loaded sol-gel coatings to enhance corrosion resistance. As the coating degrades, the PCTyr Schiff base was released slowly. Then, Mg ions enriched around the interface provided more chance for chelating with PCTyr Schiff base and drove the deposition of PCTyr Schiff base-Mg on the coating defects. Therefore, a dense protective layer was formed gradually to slow down further corrosion, which reveals the self-healing mechanism of the PCTyr Schiff base sol-gel coatings.

Heterocyclic compounds, especially ones that have a lone pair of electrons, were considered effective inhibitors in the electrolyte or sol-gel coatings through the adsorption or complexation with metal surfaces [68,69]. Shi et al. prepared a sol-gel coating with GPTMS and tetraethoxysilane (TMOS) as precursors and added 2-methylpiperidine during the gel synthesis [70]. The EIS results showed that the corrosion resistance of the sol-gel coating on AZ91D was significantly improved after adding 2-methylpiperidine. Wang et al. prepared phytic acid/silane composite coatings on AZ31 Mg alloys to improve corrosion resistance and biological activity [71]. When the molar ratio of phytic acid-to-3-aminopropyltrimethoxysilane (γ-APS) was 1:1, the hybrid coating was intact without noticeable cracks. When tested in a SBF solution, the corrosion impedance of the coated Mg alloy (13,835 Ω·cm^2^) was about 27 times higher than that of the bare substrate (507 Ω·cm^2^). The immersion test showed that with the increase of immersion time in SBF, the hybrid coating gradually dissolved due to the hydrolysis of the chemical bonds in the hybrid coating but did not peel off. In addition, the dissolution zone induces the deposition and growth of cap mineralized layers, which can provide further corrosion protection for Mg alloy. Li et al. also performed a comparable investigation and got a similar conclusion [72]. Hydroxyapatite (HA) has previously been proven as a corrosion inhibitor because of its buffering effect and reducing potential gradients between anodes and cathodes [73]. Nikbakht et al. added HA nanoparticles to the sol-gel, which helped to optimize the barrier properties of the coating [74]. The results showed that the addition of 500 mg/L to 1000 mg/L provided desirable corrosion resistance. The corrosion resistance of the modified sol-gel was three orders higher than that of the bare substrate in the SBF solution. When a higher amount of HA (2000 mg/L) was added in sol-gel, an agglomeration of nanoparticles was observed, resulting in bad corrosion resistance.

Mg^2+^ can chelate with many organic groups to form corrosion-resistant products [75,76]. Therefore, taking advantage of this feature is a good way to design efficient self-healing coatings. According to the research of various scholars, when a corrosive medium such as Cl^−^ reaches the substrate through the macro or nanopores of the sol-gel, Mg^2+,^ and corrosion inhibitors will be released. Then, Mg^2+^ can form complexes with the corrosion inhibitors to fill the macro or nanopores, resulting in a more compact sol-gel coating. Hence, the barrier effect of the coating will be well improved. The principle of sol-gel anticorrosion with corrosion inhibitors is shown in Figure 4.

### 2.2. Bare Sol-Gel with Nanoparticles

In recent years, carbon nanostructures (i.e., graphene, fullerene, nanodiamond, and carbon nanotube) have been widely used as nanofillers in composite materials due to their good functionalization ability, high mechanical strength, good electrical properties, and excellent chemical inertness [77]. It was also demonstrated that the addition of nanoparticles to the hybrid matrix might reduce the crack-forming ability and porosity of the sol-gel coatings. There are many reports on the addition of carbon nanostructures as fillers in sol-gel coatings. For instance, Nezamdoust et al. successfully deposited various amounts of hydroxylated multi-walled carbon nanotube composited into the sol-gel (PTMS/OH-MWCNT) on AM60B Mg alloy [78]. SEM observed that the micro-cracks on the pure PTMS sol-gel coating disappeared after the incorporation of OH-MWCNTs. As the nanotube content increased, the surface roughness of the sol-gel decreased, possibly due to the formation of denser, low-porosity coatings. Moreover, the corrosion resistance of the phenyl-trimethoxysilane (PTMS) sol-gel film was significantly enhanced after the addition of OH-MWCNTs at a concentration of 500 ppm. The main mechanism was due to the filling of defects and the formation of longer corrosion paths by the added OH-MWCNTs. At the same time, the water contact angle increased from approximately 86.95° to 94.65°, which indicated a significant improvement in hydrophobicity. Malik et al. prepared a GPTMS/graphene oxide (GPTMS/GO) coating on AZ91 Mg alloy by the chemical co-deposition technique [79]. The graphene oxide sheets were grafted with silanol groups. In aqueous 3.5 wt.% NaCl, the electrochemical tests showed that the corrosion resistance of AZ91 Mg alloy was improved. Since the graphene oxide-grafted GPTMS forms a passive layer on Mg alloy, a covalent metal siloxane bond (Mg-O-Si) and a layered structure of graphene oxide are formed on the substrate, which increases its hydrophobicity to 108° and enhances its adhesion and hardness.

Samadianfard et al. added sodium dodecyl sulfate-modified fullerene (F-SDS) and oxidized fullerene (OF) nanoparticles in sol-gel [80,81]. EIS experiments performed in 3.5 wt.% NaCl solution confirmed that the addition of fullerene nanoparticles significantly enhanced the corrosion resistance of the sol-gel coating. The mechanism was attributed to the decrease in the number of defects through chemical interactions. Similarly, after adding F-SDS nanoparticles, the micro-defects in the sol-gel coating also well decreased. In addition, the EIS tests revealed that the corrosion protection performance of the sol-gel coating was significantly improved after the addition of F-SDS nanoparticles (500 ppm). Nezamdoust et al. synthesized sol-gel coatings containing different amounts of hydroxylated nanodiamonds (HNDs) and deposited them on AM60B Mg alloys [82]. After adding 0.01, 0.02, and 0.05 wt.% of HND nanoparticles, the micro-defects in the sol-gel coatings were well decreased. AFM analysis showed that the average roughness of the sol-gel film was about 6.7 nm, which increased to 16.1 and 20.2 nm after adding 0.005 and 0.02 wt.% of HND, respectively. When the mass fraction of HNDs was 0.01 wt.%, the corrosion protection effect was the best. The enhanced corrosion resistance was attributed to the denseness of the coating (due to the chemical interaction with HND), the formation of tortuous paths for the diffusion of the corrosion solution, and the filling of defects by nanoparticles.

Other nanoparticle fillers (such as silica particles [83,84]) were also able to be incorporated with sol-gel coatings. Wang et al. combined the sol-gel system with fluorinated attapulgite particles (FATP@SiO_2_) to prepare a superhydrophobic surface on the AZ31 Mg alloy (ATP is an inexpensive magnesium-aluminosilicate-rich clay mineral with nanorod-like crystal morphology and reactive -OH groups on the surface) [85]. The water contact angle of the prepared surface was as high as 161° with a sliding angle of 4°. The i_corr_ value of coating was 5.519 × 10^−8^ A/cm^2^, decreased by three orders of magnitude compared to bare AZ31. The results of EIS demonstrated that the corrosion resistance of the coating decreased gradually with the prolongation of immersion time.

According to the above works, when nanoparticles are added to the silane-based sol-gel, a denser coating can be formed. The corrosive medium (such as Cl^−^) is hard to reach the substrate, thereby improving the protective effect on Mg alloys. The principle of sol-gel anticorrosion with nanoparticles is shown in Figure 5.

### 2.3. Hybrid (Inhibitors and Nanoparticles)

A possible synergistic effect can be achieved by employing both nanoparticles and inhibitors in the sol-gel coating. For example, Ashraf et al. studied the improvement of corrosion protection by introducing a series of amino acids as inhibitors and TiO_2_ nanoparticles as surface modifiers in sol-gel coatings [86]. Electrochemical results showed that a sequence of four amino acids used in the sol-gel enhanced the protection performance of the coating. The sequence of amino acids used was cysteine > serine > alanine > arginine. When the sol-gel coating containing 0.5 wt.% cysteine and 1.0 wt.% TiO_2_ nanoparticles had the best result. It effectively inhibited the corrosion of AZ91 Mg alloy in aqueous NaCl. After 2 h of immersion, the value of R_ct_ reached to 224.09 kΩ·cm^2^.

However, adding corrosion inhibitors directly into the coating matrix may deteriorate the coating or may reduce the activity of the corrosion inhibitor. Therefore, the encapsulation/intercalation of corrosion inhibitors using micro/nano-containers has proven to be a better option to provide long-term corrosion protection and self-healing capabilities. The use of natural organoclay mineral nanotubes provides an efficient and environmentally friendly method for the encapsulation of corrosion inhibitors. Montemor et al. investigated the corrosion behavior of AZ31 Mg alloy treated with multi-walled carbon nanotubes (CNTs) modified water-soluble bisaminosilane [87]. Before applying the silane solution, the carbon nanotubes were treated with an aqueous rare earth solution containing cerium nitrate or lanthanum nitrate. Electrochemical studies showed that the activation of carbon nanotubes with rare earth salts inhibited the corrosion of the Mg AZ31. Analytical and microscopic studies revealed that carbon nanotubes were uniformly dispersed in the sol-gel coating, and the carbon nanotubes acted as an inhibitor container. Adsul et al. first encapsulated the corrosion inhibitor Ce^3+^/Zr^4+^ in kaolin clay nanotubes and aluminum-pillared montmorillonite clay, and was then dispersed into the sol-gel matrix before finally being deposited on the AZ91D Mg alloy via the impregnation method [88,89]. After an immersion in 3.5% NaCl solution for 1–24 h, the self-healing and anti-corrosion abilities of the coatings were evaluated by weight-loss experiments, potentiodynamic polarization, and EIS measurements. These results confirmed that this new method had self-healing and good anticorrosion properties to AZ91D Mg alloy.

### 2.4. Substrate Pretreatment and Repair Agent

In addition to adding fillers during the preparation of sol-gel, some scholars have found that the substrate treatment (i.e., acid treatment, alkali treatment, heat treatment) will also affect the protective effect of sol-gel coatings. Saxena et al. found that specific alkali treatments of the Mg substrate before sol-gel coating can improve the corrosion resistance of the coatings because they can promote the sol-gel condensation process [90]. Supplit et al. studied the anticorrosion effect of the sol-gel coating on the AZ31 Mg alloy after the substrate was pickled with hydrofluoric acid, phosphoric acid, nitric acid, and acetic acid [91]. Pickling significantly reduced the corrosion of AZ31 Mg alloy, and acetic or hydrofluoric acid provided the best results. Hydrofluoric acid was preferred because the optical appearance of the Mg surface was better after hydrofluoric acid treatment. Dalmoro et al. studied the effect of hydrofluoric acid, acetic acid, and Na_3_PO_4_/NaOH pretreatment in the preparation of organophosphine-sol-gel hybrid coatings on AZ91 Mg alloy [92]. The analysis results showed that Na_3_PO_4_/NaOH pretreatment can form a good and stable passivation layer, which was beneficial to the further deposition of the sol-gel coating. Therefore, the alkaline pretreatment of the AZ91 alloy has advantages over pickling (acetic acid or hydrofluoric acid). L. Diaz et al. studied the effect of heat treatment (200 °C) of Mg alloy substrate on the corrosion resistance of organic-inorganic composite sol-gel coatings [93]. Heat treatment improves the protective performance of the passivation film on AZ61, thereby inhibiting the dissolution of Mg and the formation of hydrophilic groups during the coating process.

Some scholars have introduced an interesting perspective to explore sol-gel coatings—repairing sol-gel coatings through electrolytes. For example, Zhong et al. studied the addition of zinc nitrate to NaCl solution for the repair of partially damaged sol-gel coatings on Mg alloys [94]. Zinc nitrate not only prevents the development of the corrosion process but also may repair partially damaged sol-gel coatings by forming precipitates covering micron-sized cracks or defects.

The full names and abbreviations of the silane precursors used in the cited documents in Section 2 are summarized in Table 1.

## 3. Composite Sol-Gel Coatings

In recent years, composite films have received extensive attention in the field of corrosion protection of Mg alloys [95]. Due to the structural limitation of a single coating, the defects are able to form a path for the corrosive medium to penetrate the coatings. The composite coating decouples the defects on each individual coating so the formation of corrosion paths is minimized, thereby improving the protection effect. The sol-gel coating can be used as both a pretreatment layer and a sealing layer because of its unique properties.

### 3.1. Sol-Gel Coating as Pretreatment

The pre-treatment layer is crucial to the corrosion protection of the Mg alloys as it is the layer that is in contact with the substrate that provides immediate protection. The pretreatment of Mg alloy by silane-based sol-gel coating can effectively improve the corrosion resistance and the adhesion of the subsequent coating [96].

Lu et al. studied the effect of sol-gel pretreatment on the properties of the Mg-rich epoxy primer of AZ91D Mg alloy [97]. After the sol-gel pretreatment, the adhesion strength of the Mg-rich epoxy coating on AZ91D alloy showed an average increase of seven MPa, which improved the adhesion force significantly. Salt spray and electrochemical tests showed that sol-gel coating not only prevented electrolytes from penetrating into the substrate but also prevented corrosion products from diffusing outward, effectively improving the barrier effect of the coating system. Liu et al. reported a two-step procedure to introduce multifunctional anticorrosion coatings on Mg alloys [98]. The first step was to treat NaOH-activated Mg with bistriethoxysilylethane (BTSE) to immobilize a tightly crosslinked corrosion-resistant sol-gel coating (Mg-B). The second step was to treat a modified Mg-B with γ-APS to form a surface with amine function groups (Mg-B-A). According to the polarization curves of Mg-B and Mg-B-A, i_corr_ was reduced by ~68% and ~89% compared with bare Mg, respectively. Fernández–Hernán et al. used functionalized graphene nanoplatelets (COOH-GNPs) as nano reinforcers (SG/GNPs) for sol-gel (SG) coating [99]. In the first 24 h of immersion in aqueous 3.5 wt.% NaCl, the SG + SG/GNP bilayer coating structure showed the highest polarization resistance and the lowest corrosion current density. After immersion for 168 h, the coated structure showed the lowest hydrogen evolution with an almost 50% reduction compared to the uncoated substrate. The presence of COOH-GNPs increased the toughness of the coating, making it more difficult for chloride ions to reach the substrate, delaying the initiation of the corrosion process, whereas multilayer SG + SG cannot remain crack-free after heat treatment because it was too thick.

Zhang et al. successfully prepared a corrosion-resistant polymer coating with self-cleaning properties on AZ31 Mg alloy by poly(3-aminopropyl)trimethoxysilane (PAPTMS) pretreatment followed by covering with polypropylene (PP) [100]. Results indicated that the PAPTMS/PP coating surface possessed a micrometer-scaled porous spherical microstructure and super-hydrophobicity with a high-water contact angle (162 ± 3.4°) and low sliding angle (5 ± 0.6°) due to the low surface energy (10.38 mJ/m^2^). Moreover, the coating exhibited a smaller water diffusion coefficient (8.12 × 10^−10^ cm^2^/s) and water uptake volume fraction (24.5 %), demonstrating low water permeability and a good physical barrier performance. As a result, the corrosion current density of PAPTMS/PP coating exhibited approximately three orders of magnitude lower than that of the AZ31 substrate, suggesting excellent corrosion resistance. Similarly, Ahadi Parsa et al. used vinyl tri-ethoxy silane (VTES) as a pretreatment and then coated it with hydroxyapatite (HA) on the surface [101]. The composite coating also had good corrosion resistance in a similar manner. Li et al. prepared a novel catechol/lysine (CA/Lys) polymeric sol-gel coating (CA/Lys@Sol-gel) [26]. Experimental results showed that the unmodified sol-gel coating failed after only 3 days of testing, while the CA/Lys@Sol-gel provided up to 18 days of corrosion protection at aqueous 0.1 M NaCl. The main reason was that the polymerized CA/Lys was adsorbed by the sol-gel coating, and then the compactness of the coating was greatly improved by filling the micro or nano defects.

### 3.2. Sol-Gel Coating as the Surface Layer

In conversion coating, the surface film/coating is produced by chemical or electrochemical treatment. This treatment converts the surface of the metal into a thin film of metal oxides/other compounds that are chemically bonded to the surface [3,11,102]. However, most chemical conversion coatings or anodizing treatments (like plasma electrolytic oxidation (PEO) or micro-arc oxidation (MAO)) have cracks and pores, which cannot provide ideal corrosion protection effects, while sol-gel can provide functions such as sealing pores or providing biocompatibility [103,104]. This section mainly reviews the influence of the combination of chemical conversion coating, electrochemical oxidation coating and sol-gel on the anticorrosion effect of Mg alloys.

#### 3.2.1. Chemical Conversion Coating/Sol-Gel

Chemical conversion treatments are usually applied on the surface to improve the corrosion resistance of Mg alloys and improve the adhesion of coatings [105,106]. Hu et al. first prepared a layer of molybdenum acid conversion coating on Mg alloy, and then silica sol was repeatedly applied to the surface three times [107]. This sol-gel coating can cover the cracks on the molybdate conversion coating, and the formed composite coating can greatly improve the corrosion resistance of the AZ91D Mg alloy. Yue et al. used a dihydrogen phosphate solution to treat the surface of AZ31 and coated it with silane KH560 [108]. A sol-gel coating of about 2 μm was formed on Mg alloy. The improvement of the corrosion resistance of phosphating Mg AZ31 by sol-gel treatment was mainly due to the sealing effect of sol-gel on the micropores of the phosphating coating. A similar investigation was carried out by Murillo–Gutiérrez et al. [109]. Pereira et al. studied the corrosion behavior of cerium conversion coating (CeP), hybrid sol-gel coating (Hyb), and CeP-Hyb composite coating deposited on the surface of WE43 Mg alloy in brine electrolyte [105]. The results showed that the surface structure of the produced cerium conversion layer was not uniform. The CeP-Hyb composite coating improved the corrosion resistance as a result of more uniform surface morphology. Local electrochemical impedance spectroscopy experiments in the mapping mode (LEIM) experiments showed that CeP-Hyb samples exhibited a self-healing ability. The Ce in the bottom conversion layer migrated to the defect site, inhibiting the development of corrosion activity.

Nezamdoust et al. applied a composite coating consisting of Ce-V conversion coating/Ti-Zr conversion coating and sol-gel coating on AM60B Mg alloy [110,111]. Scanning electron microscopy and energy dispersive X-ray spectroscopy (XPS) analysis showed that the Ce-V conversion coating had many cracks. Similarly, a layer of cracked conversion coating was formed on Mg alloy after Ti-Zr impregnation. Then, a dense thin sol-gel film completely sealed all cracks in the conversion coating. Potentiodynamic polarization and EIS experiments in Harrison solution showed that the Ce-V conversion coating/Ti-Zr conversion coating provided limited protection against corrosion, while the composite coating significantly improved the corrosion resistance of Mg alloys. The sol-gel film provides protection against corrosion by sealing the cracks of the Ce-V conversion coating and acting as a barrier. Durán et al. mainly evaluated the effect of fluorine-based pretreatment time on the protective degradation mechanism of TEOS/GPTMS hybrid sol-gel coating [112]. The experimental results show that the long-term pretreatment was beneficial to the formation of the hydroxy magnesium fluoride layer with a higher F/O ratio, thereby improving the corrosion resistance of the coating on WE54 Mg alloy.

Ashassi–Sorkhabi et al. applied cerium lanthanum permanganate (CLP) conversion coatings on Mg alloys prior to the sol-gel process [113,114,115]. CLP coatings prevented severe corrosion of Mg in acidic sols. This pretreatment stabilized Mg and led to better adhesion of the sol-gel coating. The results showed that the coatings with corrosion inhibitors had much fewer cracks and pores after immersion in aqueous 3.5wt.% NaCl. The best-performing coating contained 1 wt.% L-aspartic acid. This observation originated from the structure of L-aspartic acid, which had three active adsorption sites, enabling its strong adsorption on the surface and providing a better anti-corrosion performance. When salt was used as a corrosion inhibitor, potassium hypophosphite showed better corrosion resistance under a short immersion time, while manganese acetate showed stronger corrosion resistance under long-term immersion. At the same time, when nanoparticles were added to the sol-gel coating as a surface modifier, the anti-corrosion effect of the coating was better. The role of nanoparticles was to increase the roughness of the coating surface, thereby reducing the number of surface pores and cracks. Guo et al. prepared Mg(OH)_2_/PMTMS/CeO_2_ hybrid coatings by a hydrothermal method [116]. The results showed that the magnesium hydroxide coating was sealed by PMTMS and CeO_2_. The thickness of the Mg(OH)_2_/PMTMS/CeO_2_ coating was about 12.86 ± 0.01 μm, which significantly improved the corrosion resistance of the AZ31 alloy.

#### 3.2.2. Anodizing/Sol-Gel

Anodizing is the process of using metals or alloys as anodes to form an oxide film on the surface via electrolysis. Bestetti et al. prepared a porous oxide layer of MgO by anodizing, followed by a single or multilayer SiO_x_ coating by the sol-gel method [117]. The anodizing of Mg improved the adhesion of the sol-gel layer, and the sol-gel layer sealed the pores of MgO. Hence, the corrosion resistance of Mg was well-improved. Lamaka et al. also performed a similar investigation [118]. While Afsharimani et al. prepared an anodizing/sol-gel coating, they added graphene nanosheets to the sol-gel to improve the corrosion resistance of Mg alloys in 0.05 M NaCl [119]. The corrosion performance of the sol-gel coating containing graphene nanoplatelets (i_corr_ = 0.01 μA/cm^2^) was better than that of the anodized coating without graphene nanoplatelets (i_corr_ = 1.00 μA/cm^2^), which was due to the better coating quality and barrier properties.

PEO is a popular process for forming porous ceramic oxide layers on metal substrates. Although PEO can solve the problem of insufficient adhesion of organic coatings and form a harder, thicker, and stronger coating, the pores and microcracks in the PEO coating structure reduce its corrosion resistance. The coating porosity is of vital concern because the interconnected pores can form direct pathways connecting the Mg alloy substrate surface with aggressive media, reducing the protective barrier effect of the PEO coating [120]. To solve this problem, combining PEO with a sol-gel coating allows the pores in the coating to be sealed by the sol-gel, thereby improving corrosion protection performance.

Ivanou et al. prepared an inhibitor-loaded PEO layer with a TiO_2_-doped sol-gel coating to the ZE41 Mg alloy [121]. The scanning vibrating electrode technique (SVET) test results showed that the composite coating had an effective anti-corrosion performance. During immersion in 50 mM of NaCl solution, the corrosion rate was reduced by a magnitude of 3 to 100 times. In this composite self-healing coating, a thin, porous PEO layer can be successfully used as a reservoir for corrosion inhibitors, in addition to providing barrier protection. In this case, the corrosion inhibitor was pinned to the metal substrate where corrosion begins, and a thin sol-gel coating on top of the PEO layer slowed its leaching rate. Similar studies were also done by Shang et al. [122], Cui et al. [123], Pezzato et al. [124], Chen et al. [76], Merino et al. [125], and Chen et al. [126]. The difference is that these scholars loaded corrosion inhibitors or nanoparticles in the porous PEO layer or in the sol-gel, which can get a better protective effect. Those works came to the same conclusion: the sol-gel layer can effectively reduce the porosity of PEO coatings and form a dense hydrophobic outer layer. The hydrophobic properties of the composite coating may be related to the siloxane network (Si-O-Si) formed on the surface. After the silane-based sol-gel treatment, the corrosion resistance of the PEO-based coating was improved significantly. 

Jiang et al. prepared N-doped graphene quantum dots (N-GQDs)/PMTMS composite sol-gel coating on AZ91D Mg alloy [127]. Compared with the bare Mg alloy, the N-GQDs/PMTMS coating showed a significant enhancement in its corrosion resistance. According to the EIS test, the value of R_ct_ was more than three orders of magnitude than that of bare Mg alloy. The mechanism was mainly due to the strong chemical bonding between the N-GQDs coating and the sealing layer of the PMTMS sol-gel.

It has been well addressed in the literature that when the chemical conversion coating and anodizing are used together with the silane-based sol-gel coating, the sol-gel can fill the defects and cracks in the coating beneath the sol-gel. The composite coating becomes denser, blocking the possible path from the aggressive external medium to the substrate. The corrosion factors, such as Cl^−^, do not easily reach the substrate, thereby improving the protection of the Mg alloy. This paper summarizes the anti-corrosion principle of the composite sol-gel coating, as shown in Figure 6.

### 3.3. Multilayer Hybrid Coating

As reported in the previous sections, the sol-gel layer can be used as both a pretreatment layer and a surface-sealing layer. However, when applied as a pretreatment layer, the sol-gel coating sometimes needs to be hydrolyzed and condensed under acidic conditions. When directly coated on the surface, the Mg alloy will corrode and generate a small amount of hydrogen, consequently resulting in more defects and poor adhesion. When used as a surface layer, sol-gel coatings are cracked due to the rapid evaporation of residual water and solvents during heat treatment. To overcome the above problems, some researchers proposed the application of multilayer sol-gel coatings. The first layer is a chemical or electrochemical conversion film. The application of a second sol-gel layer provides a chemically neutral surface that facilitates the addition of another thick coating. 

Toorani et al. studied morphology, surface properties, and corrosion resistance of PEO/sol-gel/epoxy three-layer coatings [128,129]. The presence of APTES in the sol-gel resulted in the formation of an amine-rich surface. The amino functional groups act as a molecular bridge that enhance the adhesion between the organic and PEO coatings. Additionally, the amino groups form a new reaction site, which is a strong nucleophile that readily reacts with the epoxy group through a ring-opening reaction. Each amino group can undergo a ring-opening reaction with an epoxy group to form a covalent bond between the two layers. There is also the possibility of hydrogen bonding between the silane layer and the epoxy layer. The study on the protective properties of epoxy coatings showed that the addition of a sol-gel layer between PEO and epoxy coatings can improve the corrosion resistance of the coating system. In addition, the authors provided evidence that the presence of cerium nitrate in the PEO coating and the presence of 8-HQ in the sol-gel layer gave the coating system better protective performance.

Wang et al. prepared a (APS/Bis [3-(triethoxysilyl)propyl]tetrasulfide (BTESPT))/graphene/(APS/BTESPT) three-layer composite coating [130]. Compared with bare AZ31B Mg alloy, the corrosion current density of the composite coating in 3.5 wt.% NaCl solution was reduced by four orders of magnitude. This was due to the excellent corrosion resistance and strong adhesion of the sol-gel coating on the surface of the Mg alloy, as well as the barrier of graphene oxide to the permeation path of corrosive media, such as Cl^−^ and H_2_O. The results indicated that the composite coating still had good corrosion resistance after immersion in 3.5 wt.% NaCl solution for 7 days.

The full names and abbreviations of the sol-gel precursors used in the cited documents in Section 3 are summarized in Table 2.

## 4. Conclusions and Outlooks

Due to the various industrial applications of Mg alloys, much research has been done to improve its anti-corrosion performance. Over the past few decades, the sol-gel methods have proven to be a good way to increase corrosion protection on Mg alloys. In addition to its own good barrier function, the versatile sol-gel coating can be modified by many approaches, which were divided into four categories in this paper: (1) Bare sol-gel; (2) Nanoparticles; (3) Corrosion inhibitors; and (4) Sol-gel-based composite coatings. All the above methods have been proven to significantly improve the barrier and protection effects of the sol-gel coatings.

In this paper, the existing modification methods of silane-based sol-gel coatings were reviewed. The sol-gel precursors of the cited article were summarized in Table 1 and Table 2. It can be seen that TEOS and GPTMS are the most commonly used silane precursors. The thickness and corrosion resistance of different sol-gel coatings in the cited article was included in Table 3.

As concluded in the present works, the sol-gel coating has a good protective effect on Mg alloys. However, there are still some issues that need further study including:Most of the synthetic methods reviewed in this paper were carried out under laboratory conditions.The durability of the coated surface is considered to be the most important aspect that should be further enhanced in future work. Although the corrosion inhibitor/nano-filler silane hybrid coating has improved its protective effect, it is far from enough to be used in the industry. The sol-gel composite coating with long-lasting corrosion protection should be addressed in a future study.There is a lack of work considering the mechanical properties of the coating, such as ductility and hardness. These properties are worth paying attention to in the practical application of coatings in industrial applications.The versatility of the coating is also very important. Apart from the anti-corrosion aspect, sol-gel coatings also need to provide oxidation resistance, abrasion resistance, water resistance, biocompatibility, and many other useful properties. With the in-depth study of sol-gel technology and related characterization techniques, sol-gel coatings will have wider and more practical applications.

## Figures and Tables

**Figure 1 molecules-28-02563-f001:**
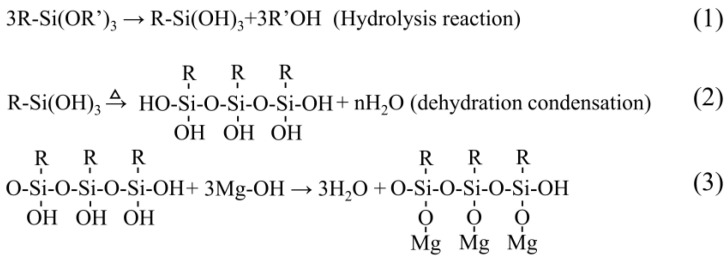
Schematic of silane coatings formation process. The first stage, siloxane hydrolysis: (1); In the second stage, dehydration condensation between silanol molecules: (2); In the third stage, the Si-OH bond is combined with the -OH on the Mg alloy: (3).

**Figure 2 molecules-28-02563-f002:**
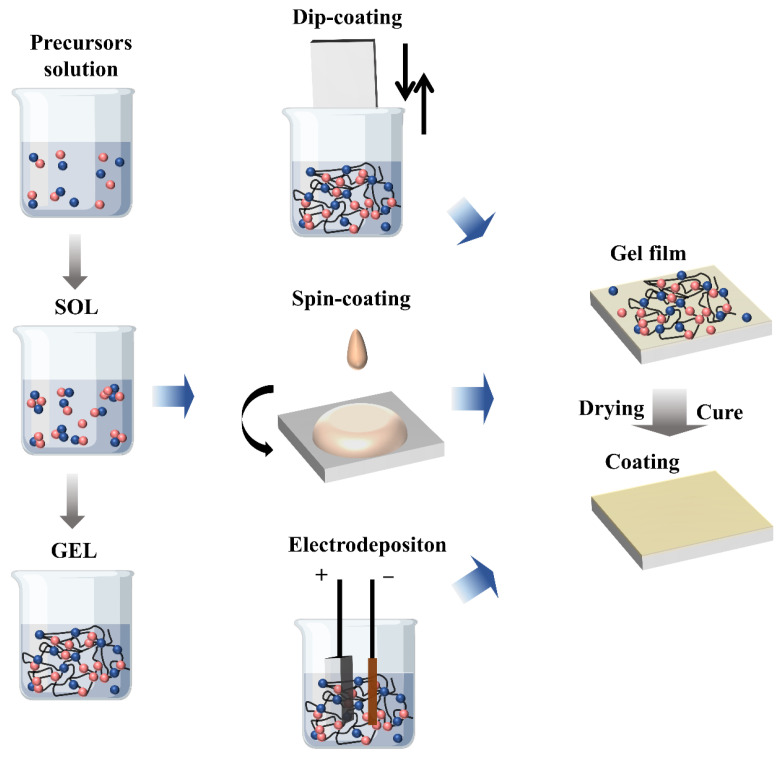
Schematic of steps and processes used to obtain sol-gel coatings.

**Figure 3 molecules-28-02563-f003:**
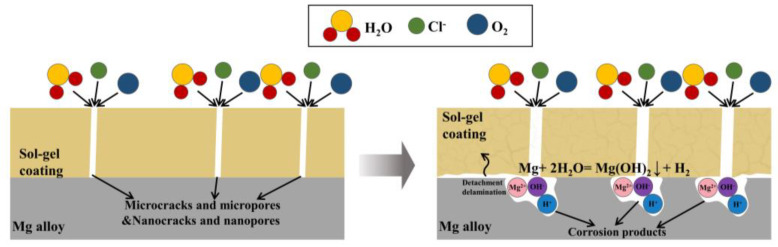
The schematic diagram of the anti-corrosion principle of bare sol-gel coatings.

**Figure 4 molecules-28-02563-f004:**
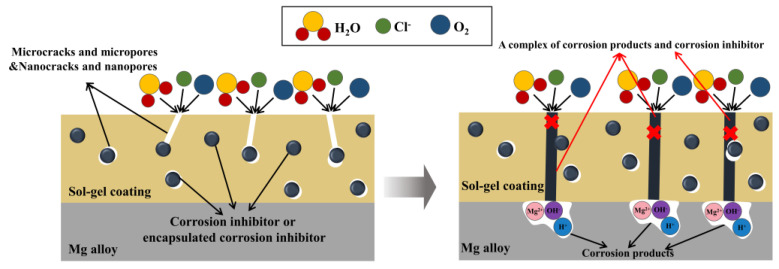
The schematic diagram of sol-gel coating containing corrosion inhibitors.

**Figure 5 molecules-28-02563-f005:**
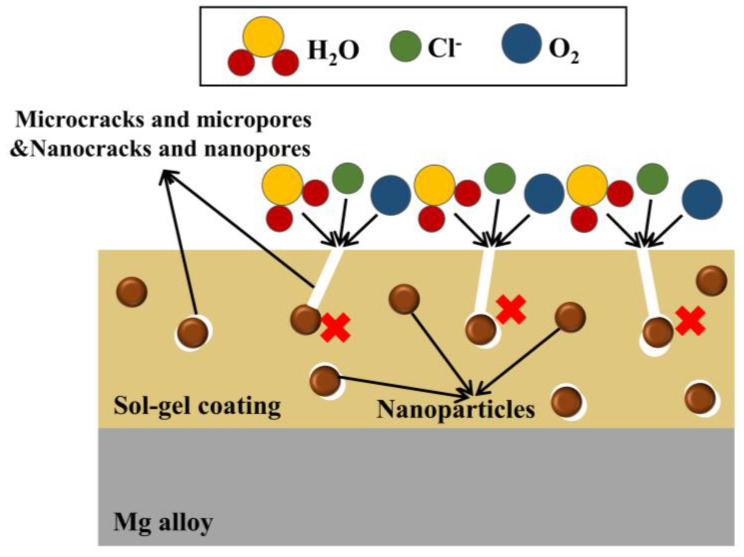
The schematic diagram of sol-gel coatings containing nanoparticles.

**Figure 6 molecules-28-02563-f006:**
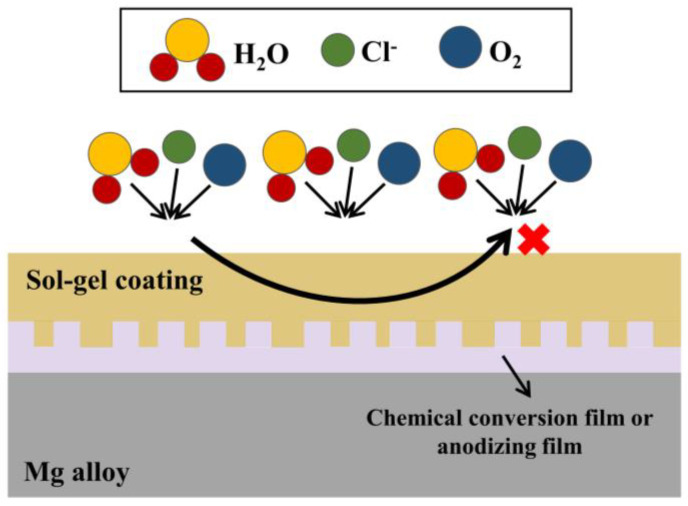
The schematic diagram of the anticorrosion principle of the composite sol-gel coatings.

**Table 1 molecules-28-02563-t001:** The full names and abbreviations of the silane precursors used in the cited literature in Section 2.

	Chemical Name	Abbreviation	Ref.
1	Tetraethoxysilane	TEOS	[37,56,57,58,63,67,74,80,81,82,85,86,88,89,91,92,94]
2	Triethoxyvinylsilane	VTEO	[56]
3	Diethylphosphonatoethyltriethoxy-silane	PHS	[57]
4	3-Glycidoxypropyltrimethoxysilane	GPTMS	[37,58,61,67,70,74,79,80,81,82,85,88,89,94]
5	Vinyltriethoxysilane	VETO	[63]
6	3-(Trimethoxysilyl)propylmethacrylate	MAP	[64]
7	Methyltriethoxysilane	MTES	[74,91]
8	Tetraethoxysilane	TMOS	[70,93]
9	3-Aminopropyltrimethoxysilane	γ-APS	[71,72]
10	Phenyl-trimethoxysilane	PTMS	[78]
11	Triethoxyvinylsilane	TEVS	[86]
12	Bis-[triethoxy amino] silane	BAS	[87]
13	Bis-1,2-(TriethoxySilyl)Ethane	BTSE	[90]
14	Methylmethoxysilane	MTMS	[92]
15	γ-Methacryloyloxypropyltrimethoxysilane	MAPTMS	[93]

**Table 2 molecules-28-02563-t002:** The full names and abbreviations of the silane precursors used in the literature cited in Section 3.

	Chemical Name	Abbreviation	Ref.
1	γ-Glycidoxy propyl trimethoxy silane	γ-GPS	[97,126]
2	Bistriethoxysilylethane	BTSE	[98]
3	3-Amino-propyltrimethox-ysilane	γ-APS	[98,129,130]
4	Poly(3-aminopropyl)trimethoxysilane	PAPTMS	[100]
5	Vinyl tri-ethoxy silane	VTES	[101]
6	Tetraethoxysilane	TEOS	[26,99,107,110,111,112,113,114,115,117,119,122,124,125,126,128,129,131,132]
7	3-Glycidoxypropyltrimethoxysilane	GPTMS	[26,76,107,108,109,110,111,112,118,119,121,125,131]
8	Phenyl-trimethoxysilane	PTMS	[110,121]
9	3-Methacryloxypropyl trimethoxysilane	-	[117]
10	Methyltriethoxysilane	MTES	[99,113,114,115,124,132]
11	γ-Amino propyltriethoxysilane	APTES	[128]
12	Bis [3-(triethoxysilyl)propyl]tetrasulfide	BTESPT	[130]
	Methylmethoxysilane	MTMS	[116,123,127]

**Table 3 molecules-28-02563-t003:** Thickness and corrosion resistance of the different sol-gel coatings.

	Substrate	Coatings	Sol-Gel Solution Composition	Thickness	Electrolyte	Anti-Corrosion Effect *	Ref.
2008	AZ91D	Bare sol-gel coating	SiO_2_ (TV) sol: the molar ratio of TEOS:VTEO: ethanol: water: acetic acid is equal to 0.25:0.75:10:4:0.01. SiO_2_ (T) sol: the molar ratio of TEOS: ethanol: water: acetic acid is equal to 1:10:4:0.01.	-	3.5 wt.% NaCl	i_corr_ Mg = 1.29 × 10^−5^ A/cm^2 ^ i_corr_ SiO_2_ (T) = 1.15 × 10^−6^ A/cm^2 ^ i_corr_ SiO_2_ (TV) = 2 × 10^−6.8^ A/cm^2^	[56]
2006	AZ31B	Bare sol-gel coating	The molar ratio of the silanes was 1:2 (PHS: TEOS).	600–700 nm	Harrison’s solution (0.35 wt.% (NH_4_)_2_SO_4_ and 0.05 wt.% NaCl)	|Z|_0.01Hz_: uncoated≪ silica coated ≪coated with PHS: TEOS film.	[57]
2017	AZ31	Bare sol-gel coating	Mixing TEOS and GPTMS precursors in a molar ratio of 3:1 and using ethanol and acetic acid as solvent and catalyst, respectively.	0.7–2.5 μm	0.1 M NaCl	i_corr_ is about an order of magnitude lower compared to magnesium alloys, simultaneously, a protection range up to 150 mV.	[58]
2010	AZ91D	Bare sol-gel coating (Ce^3+^)	Mixing GPTMS, VETO, distilled water, and ethanol in 1:3:12:30 molar ratios. Ce(NO_3_)_3_·6H_2_O was added to yield 0.01 M of Ce^3+^.	-	3.5 wt.% NaCl	i_corr_ Mg = 1.29 × 10^−5^ A/cm^2 ^ i_corr_ sol-gel = 8.64 × 10^−7^ A/cm^2 ^ i_corr_ sol-gel (MPD) = 5.75 × 10^−8^ A/cm^2^	[63]
2015	Elektron 21 (El21) alloy	Bare sol-gel coating (Ce^3+^)	Mixing the starting precursors consisting of TEOS and MAP, deionized water, and ethanol with a molar ratio of 11:1:60:80. The production of cerium-doped sols was performed by adding cerium nitrate (Ce(NO_3_)_3_ · 6 H_2_O) at four different concentrations: (0.005, 0.01, 0.05, and 0.1) mol/L.	1 μm	0.05 M NaCl	The hybrid film exhibited a high resistive modulus (10^5^–10^6^ Ω cm^2^) during the first few hours of immersion, and the addition of cerium at a concentration of 0.01 M to the sol significantly increased the durability of the film (2 days).	[64]
2020	AZ31	Bare sol-gel coating (Ce^3+^)	Mixing TEOS and GPTMS in a molar ratio of 3:1 that was dissolved in ethanol. Then use an inhibitor solution of 2.5 mol% Ce(NO_3_)_3_ (relative to the precursor) and a catalyst of 2.5 vol% AcOH. Obtain mixed sols by mixing the two solutions at a volume ratio of 4.5:1. The hybrid sol was obtained by mixing both solutions in a volume ratio of 4.5:1.	0.9–3.3 μm	0.1 M NaCl	Hybrid coatings achieved a reduction of the corrosion current density by about two and three orders of magnitude with regard to the undoped coated specimen and the AZ31 alloy respectively, also exhibiting a protection range of up to 160 mV.	[37]
2010	AZ31	Bare sol-gel coating (8-HQ)	Solution A: zirconium (IV) propoxide (70% solution in 2-propanol) and ethylacetoacetate with volume ratio 1:1. Solution B: GPTMS and 2-propanol with 1:1 volume ratio. The final solution: solutions (A + B) with a volume ratio of 1:1. Inhibitor-doped sol-gel films were prepared adding 0.26 wt.% of 8-HQ.	3 μm	0.005 M NaCl	After 14 days immersed, |Z|_0.01Hz_ sol-gel ≈2 × 10^5^ Ω cm^2^ |Z|_0.01Hz_ sol-gel(8-HQ)≈ 1 MΩ cm^2 ^ R_ct_ sol-gel = 687 kΩ cm^2^ R_ct_ sol-gel (8-HQ) = 1649 kΩ cm^2^	[61]
2010	ZE21B	Sol-gel coating + corrosion inhibitor	Mixing GPTMS and TEOS (molar ratios = 5:1), the solvent is an appropriate amount of distilled water and ethanol. The above solution was doped with corrosion inhibitor (PCTyr Schiff base).	-	Simulated Body Fluid (SBF)	i_corr_ Mg = 1.31 × 10^−4^ A/cm^2 ^ i_corr_ sol-gel = 4.29 × 10^−6^ A/cm^2 ^ i_corr_ sol-gel (PCTyr) = 3.64 × 10^−6^ A/cm^2^	[67]
2009	AZ91D	Sol-gel coating + corrosion inhibitor	The mixed precursors were GPTMS and TMOS with molar ratio of 3:1 in acetic acid solution of 0.05 mol/L. The molar ratio of GPTMS: acetic acid is 60:1. The above solution was doped with 0.001 mol/L MPD. The inhibitor was pre-resolved in 20 mL of distilled water and then added into sol solution.	-	Harrison’s solution	i_corr_ Mg = 7.10 × 10^−3^ A/cm^2^ i_corr_ sol-gel = 2.41 × 10^−7^ A/cm^2^ i_corr_ sol-gel (MPD) = 4.5 ×10^−10^ A/cm^2^	[70]
2017	AZ31	Sol-gel coating + corrosion inhibitor	Phytic acid and γ-APS (mole ratios were 1:1) were added into 40 mL of mixed solution with water/ethanol volume ratio of 3:2.	-	SBF	i_corr_ Mg = 49.41 μA/cm^2 ^ i_corr_ sol-gel (Phytic acid) = 3.57 μA/cm^2^	[71]
2021	AZ31	Sol-gel coating + corrosion inhibitor	The silane sols consisted of three different precursors: MTES, GPTMS, and TEOS in equal volumes (6.6% *V/V*) in a combination of 10% distilled water and 70% ethanol. 1000 mg/L hydroxyapatite (HA) nanoparticles were added to the sol.	3.81 μm	SBF solution	R_f_: the overall resistance of the coating response. The R_f_ of silane coating modified with HA nanoparticles reached 41 kΩ cm^2^, which was more than 100 times higher than that without modification after being soaked for 4 days.	[74]
2018	AM60B	Sol-gel coating + nanoparticles	500 ppm OH-MWCNTs were added to PTMS, and the mixture was ultrasonically agitated for about 20 min.	1.4–1.5 μm	Harrison’s solution	R_p_ sol-gel = 207.5 kΩ cm^2 ^ R_p_ sol-gel (OH-MWCNTs) = 368.6 kΩ cm^2 ^ 1440min later. R_p_ sol-gel = 22.6 kΩ cm^2 ^ R_p_ sol-gel (OH-MWCNTs) = 44.1 kΩ cm^2^	[78]
2021	AZ91	Sol-gel coating + nanoparticles	The GPTMS/GO was prepared by mixing 0.25 mL GO, 10 mL ethanol, 10 mL GPTMS, and 79.85 mL deionized water.	10 μm	3.5 wt.% NaCl	i_corr_ Mg = 49.90 μA/cm^2 ^ i_corr_ sol-gel = 0.25 μA/cm^2 ^ i_corr_ sol-gel (GO) = 0.016 μA/cm^2 ^ R_ct_ Mg = 0.87 kΩ cm^2 ^ R_ct_ sol-gel = 3.9 kΩ cm^2 ^ R_ct_ sol-gel (GO) =5.02 kΩ cm^2^	[79]
2020	AM60B	Sol-gel coating + nanoparticles	Mixing 0.02 mol TEOS, 0.02 mol GPTMS 0.14 mol water (the pH was formulated to 1.5 with HCl). Then, 100 mg/L oxidized fullerene was added into the sol.	1.5–2 μm	3.5 wt.% NaCl	after being soaked for 1440 min. R_p_ Mg = 1.405 kΩ cm^2 ^ R_p_ sol-gel (OF) = 500.018 kΩ cm^2^	[80]
2021	AM60B	Sol-gel coating + nanoparticles	Mixing 0.02 mol TEOS, 0.02 mol GPTMS 0.14 mol water (the pH was formulated to 1.5 with HCl). Then, 500 ppm F-SDS (the SDS molecules were stabilized on the fullerene C60 nanoparticles) was added into the sol.	3 μm	3.5 wt.% NaCl	After being soaked for 48h. R_p_ sol-gel = 6 kΩ cm^2^ R_p_ sol-gel (F-SDS) = 23 kΩ cm^2^	[81]
2020	AM60B	Sol-gel coating + nanoparticles	Mixing 0.02 mol TEOS and 0.02 mol GPTMS. Acidic water (pH = 1, HCl) was added to the sol with 1:1 alkoxy to H_2_O molar ratio. Then, 0.01 wt.% of the hydroxylated nanodiamonds was added into the sol.	0.7–0.8 μm	Harrison’s solution	After 3 h immersion, i_corr_ sol-gel = 2.202 μA/cm^2 ^ i_corr_ sol-gel (HND) = 0.476 μA/cm^2^	[82]
2020	AZ31B	Sol-gel coating + nanoparticles	The hydrolysis and polymerization of GPTMS and TEOS were under acidic condition. After a certain amount of F-ATP@SiO2 particles were added to the sol.	-	3.5 wt.% NaCl	After 3 h immersion, i_corr_ Mg = 7.143 × 10^−5^ A/cm^2 ^ i_corr_ sol-gel (SiO_2_) = 5.519 × 10^−8^ μA/cm^2^	[85]
2019	AZ91	Sol-gel coating + inhibitor + nanoparticles	Adding TEOS and TEVS with a molar ratio of 1:3. 0.5 wt.% cysteine and 1.0 wt.% TiO_2_ were added to the sol.	450 nm	0.05 M NaCl	i_corr_ sol-gel = 1168.1 nA/cm^2 ^ i_corr_ sol-gel (cysteine +TiO_2_) = 25.0 nA/cm^2^ R_ct_ Mg = 0.238 kΩ cm^2 ^ R_ct_ sol-gel = 5.554 kΩ cm^2 ^ R_ct_ sol-gel (cysteine +TiO_2_) = 224.090 kΩ cm^2^	[86]
2008	AZ31	Sol-gel coating + inhibitor + nanoparticles	The BAS was prepared by dissolving 5% (vol/vol) of silane in a mixture of methanol (10% vol/vol) and 85% (vol/vol) of distilled water. Modified by Ce(NO_3_)_3_ or La(NO_3_)_3_ CNTs were then added to the sol.	5.5–6 μm	0.05 M NaCl	For the silane coating modified with the untreated CNTs, the anodic current densities attained values around 60 µA/cm^2^ for all the test period (up to 24 h) and the cathodic currents attained values around −60 µA/cm^2^. The silane coatings modified with the CNTs treated with cerium revealed the lowest corrosion activity. After 24 h of immersion, the activity decreased and both anodic and cathodic current densities ranged between 2 and −2 μA/cm^2^.	[87]
2017	AZ91D	Sol-gel coating + inhibitor + nanoparticles	The organic-inorganic hybrid matrix sol was prepared by hydrolysis of GPTMS with TEOS in molar ratio of 3:5:1 with 0.1 N HCl as catalyst.	-	3.5 wt.% NaCl	After being soaked for 24h, i_corr_ Mg =6.0 × 10^−5^ A/cm^2^ i_corr_ sol-gel = 1.4 × 10^−6^ A/cm^2^ i_corr_ sol-gel (Ce^3+^/Zr^4^ +halloysite nanotubes) = 0.9 × 10^−6^ A/cm^2^	[88]
2018	AZ91D	Sol-gel coating + inhibitor + nanoparticles	GPTMS and TEOS were taken in molar ratio of 3:5 and hydrolysed in presence of 0.1 N HCl as catalyst, to synthesize the hybrid organic-inorganic matrix sol.	2.5 ± 0.5 μm	3.5 wt.% NaCl	After being soaked for 120 h, i_corr_ sol-gel = 1.513 × 10^−5^ A/cm^2^ i_corr_ sol-gel (Ce^3+^/Zr^4^ +halloysite nanotubes) = 5.602 × 10^−7^ A/cm^2^	[89]
2010	AZ91D	Bare sol-gel coating	The silane sol was synthesized by mixing GPTMS, TEOS, distilled water, and ethanol in 3:1:13:40 molar ratios.	-	0.005 M NaCl + zinc nitrate	After the introduction of zinc nitrate for 48 h, the resistance value increased and arrived at about 180 kΩ, which exceeded that of the sample initially immersed in undoped solution for 1 h (about 140 kΩ).	[94]
2012	AZ91D	Composite sol-gel coating (sol-gel/ Mg-rich epoxy primer)	Adding 10 wt.% γ-GPS to a 1:8 mixture of methanol and distilled water. Glycerol (0.15 vol.% of the total silane solution) was added.	-	3 wt.% NaCl	|Z|_0.01Hz_ sol-gel/ Mg-rich epoxy primer: higher than 10^11^ Ω cm^2^; For magnesium-rich primer of AZ91D alloy without pre-treatment, EIS results show that the alloy substrate is corroded after 840 h immersion; for magnesium rich primer of AZ91D alloy pretreated with silane, the EIS results show that the substrate will corrode after being soaked for 1800 h.	[97]
2013	AZ31	Composite sol-gel coating (sol-gel/ sol-gel)	BTSE or γ-APS solution was prepared by mixing 5% silane, 90% ethanol, and 5% Milli-Q water.	-	SBF solution	i_corr_ Mg = 8.32 ± 0.63 μA/cm^2^ i_corr_ sol-gel = 2.69 ± 0.31 μA/cm^2^ i_corr_ sol-gel /sol-gel = 0.90 ± 0.24 μA/cm^2^ R_p_ Mg = 2650 ± 538 Ω cm^2^ R_p_ Mg-B = 7788 ± 2572 Ω cm^2^ R_p_ Mg-B-A =13635 ± 2745 Ω cm^2^	[98]
2020	AZ31B	Single sol-gel coating + nanoparticles	TEOS + MTES/Isopropanol/water: 1/5/10. To obtain the initial sols, TEOS and MTES were mixed in molar fraction of 40 %/60 %. 0.005 wt.% COOH-GNPs was added to isopropyl alcohol, and the final concentration of nano charges measured in the coating was 0.046 wt.% of COOH-GNPs.	2.2 μm	3.5 wt.% NaCl	After being soaked for 24 h, i_corr_ Mg = 6.6 × 10^−6^ A/cm^2^ i_corr_ sol-gel =1.2 × 10^−6^ A/cm^2^ i_corr_ sol-gel /sol-gel = 5.2 × 10^−7^ A/cm^2^	[99]
2020	AZ31	Composite sol-gel coating (sol-gel/ PP)	PAPTMS: (APTMS: ethanol: deionized water = 3:22:75, *V/V/V*)	About 60 μm	3.5 wt.% NaCl	i_corr_ Mg = 4.96 × 10^−5^ A/cm^2^ i_corr_ sol-gel = 1.95 × 10^−6^ A/cm^2^ i_corr_ sol-gel/ PP = 9.08 × 10^−8^ A/cm^2^ R_p_ Mg = 190.9 Ω cm^2^ R_p_ sol-gel = 7578 Ω cm^2^ R_p_ sol-gel/ PP = 2.80 × 10^5^ Ω cm^2^	[100]
2022	AZ31	Composite sol-gel coating (sol-gel/ HA)	2.5 mL of VTES was added to a mixture of 5 mL DIW, 5 mL of acetone, and 95 mL of ethanol under magnetic stirring.	-	3.5 wt.% NaCl	i_corr_ Mg= 36.1 ± 0.1 μA/cm^2^ i_corr_ sol-gel/ HA = 0.9 ± 0.1 μA/cm^2^ R_p_ Mg = 253 Ω cm^2^ R_p_ sol-gel/ HA = 12155 Ω cm^2^	[101]
2023	AZ31	Composite sol-gel coating (sol-gel/(CA/Lys))	Silicon sol was prepared from GPTMS, TEOS, deionized water, and ethanol, mixed in a volume ratio of 3:1:1:5. Then, add Ce(NO_3_)_3_ to make the concentration of Ce(NO_3_)_3_ reach 0.01 M.	9 ± 0.5 μm	0.1 M NaCl	Sol-gel: after being soaked for 4 days, R_ct_ = 1.030e^3^. CA/Lys @Sol-gel: after being soaked for 4day, R_ct_ = 1.344e^6^. After 18 days of the test, the value of R_ct_ was still as high as 10^5^ Ohm⋅cm^2^.	[26]
2009	AZ91D	Composite sol-gel coating (molybdate/sol-gel)	Silicon sol was prepared from TEOS, GPTMS, and ethanol, which were mixed in a molar ratio of 0.25:0.75:10.	-	3.5 wt.% NaCl	i_corr_ Mg= 1.29 × 10^−5^ A/cm^2^ i_corr_ conversion coating = 1.76 × 10^−5^ A/cm^2^ i_corr_ conversion coating/sol-gel = 3.80 × 10^−5^ A/cm^2^ R_p_ conversion coating = 552 Ω cm^2^ R_p_ conversion coating/sol-gel = 4.5× 10^4^ Ω cm^2^	[107]
2013	Elektron21	Composite sol-gel coating (phosphate /sol-gel)	Mixing starting precursors consisting of GPTMS and aluminum-tri-sec-butoxide, deionized water, and propanol in a molar ratio of 2:1:1:10.	7 μm	0.05 M NaCl	After being soaked for 192 h, |Z|_0.01Hz_ Mg ≈3 × 10^3^ Ω cm^2^ |Z|_0.01Hz_ sol-gel≈3 × 10^3^ Ω cm^2^ |Z|_0.01Hz_ conversion coating /sol-gel ≈ 3 × 10^4^ Ω cm^2^	[109]
2013	AZ31	Composite sol-gel coating (phosphate /sol-gel)	Silane solution: 10 g/L KH560	4.9 μm	3.5 wt.% NaCl	R_tot_ conversion coating = 2227 Ω cm^2^ R_tot_ conversion coating/sol-gel = 5.6 × 10^3^ Ω cm^2^	[108]
2022	WE43	Composite sol-gel coating (Cerium/sol-gel)	The inorganic TEOS (10% *V/V*) and organic GPTMS (20% *V/V*) precursors were added together to a mixture of ethanol (10% *V/V*) and distilled water (60% *V/V*).	2.06 ± 0.05 μm	0.1 M NaCl	After being soaked for 24 h, i_corr_ Mg = 10.9 μA/cm^2^ i_corr_ conversion coating = 3.0 μA/cm^2^ i_corr_ conversion coating/sol-gel = 0.6 μA/cm^2^ R_ct_ Mg = 3177 Ω cm^2^ R_ct_ conversion coating = 4363 Ω cm^2^ R_ct_ conversion coating /sol-gel = 22485 Ω cm^2^	[131]
2017	AM60B	Composite sol-gel coating (Cerium vanadate /sol-gel)	Mixing 0.04 mol TEOS, 0.02 mol GPTMS, and 1.23 mol acidic water so that the molar ratio of the water molecules to alkoxide groups was about 5:1.	2 μm	Harrison’s solution	i_corr_ Mg = 310.9 μA/cm^2^ i_corr_ conversion coating = 145.8 μA/cm^2^ i_corr_ conversion coating/sol-gel = 4.6 μA/cm^2^ R_ct_ Mg =21.4 Ω cm^2^ R_ct_ conversion coating = 114.5 Ω cm^2^ R_ct_ conversion coating /sol-gel = 3750.0 Ω cm^2^	[111]
2019	AM60B	Composite sol-gel coating (Ti-Zr/sol-gel)	Here, 0.02 mol TEOS and 0.02 mol GPTMS precursors were mixed.	1.5–2 μm	0.05 M NaCl	i_corr_ Mg= 9.670 μA/cm^2^ i_corr_ conversion coating = 5.692 μA/cm^2^ i_corr_ conversion coating/sol-gel = 0.027 μA/cm^2^ R_p_ Mg = 4.7× 10^3^ Ω cm^2^ R_p_ conversion coating = 7.9× 10^3^ Ω cm^2^ R_p_ conversion coating/sol-gel = 858.5 × 10^3^ Ω cm^2^	[110]
2021	WE54	Composite sol-gel coating (fluoride/sol-gel)	Hybrid sols were synthesized by mixing TEOS and GPTMS in a molar ratio of 3:1, employing ethanol as solvent and an acidic mixture of acetic acid and nitric acid as catalysts in a volume proportion of 2.5:1.	-	0.1 M NaCl	i_corr_ Mg = 1.78 × 10^−5^ A/cm^2^ i_corr_ conversion coating = 1.84 × 10^−6^ A/cm^2^ i_corr_ conversion coating/sol-gel = 1.86 × 10^−7^ A/cm^2^	[112]
2019	AZ91	Composite sol-gel coating (CLP/sol-gel)	Adding TEOS and MTES with a molar ratio of 2:3 to an acidic solution of nitric and acetic acids in 1:5 vol ratio. Then, 1 wt.% L-Aspartic was added to the sol as corrosion inhibitors.	850 nm	3.5% NaCl	R_total_ Mg = 0.119 kΩ cm^2^ R_total_ conversion coating =0.681 kΩ cm^2^ R_total_ conversion coating /sol-gel= 85.417 kΩ cm^2^	[113]
2019	AZ91	Composite sol-gel coating (CLP/sol-gel)	A mixture of TEOS and MTES with a molar ratio of 2:3 was hydrolyzed in a solution of acetic and nitric acids in a 5:1 vol ratio. Then, 0.5 wt.% of cloisite Na^+^ and 0.5 wt.% of methionine were added to the sol.	800 nm	3.5% NaCl	R_total_ Mg = 0.119 kΩ cm^2^ R_total_ conversion coating =0.681 kΩ cm^2^ R_total_ conversion coating /sol-gel= 434.731 kΩ cm^2^	[114]
2019	AZ91	Composite sol-gel coating (CLP/sol-gel)	A mixture of TEOS and MTES with a molar ratio of 2:3 was hydrolyzed in a solution of acetic and nitric acids in a 5:1 vol ratio. Then, 0.5 wt.% potassium hypophosphite and 0.5 wt.% of cloisite 20A nanoparticle were added to the sol.	-	3.5% NaCl	R_total_ Mg = 0.119 kΩ cm^2^ R_total_ conversion coating = 0.681 kΩ cm^2^ R_total_ conversion coating /sol-gel = 127.382 kΩ cm^2^	[115]
2017	AZ31	Composite sol-gel coating (Mg (OH)_2_/sol-gel)	PMTMS/CeO2: a mixture of MTMS, ethanol and water (3:10:20, *V/V/V*), cerium nitrate (10^−3^ M).	12.86 ± 0.01 μm	3.5 wt.% NaCl	i_corr_ Mg = 1.51 ± 0.08 × 10^−5^ A/cm^2^ i_corr_ Mg(OH)_2_/sol-gel = 2.46 ± 0.07 × 10^−8^ A/cm^2^ R_ct_ Mg = 854.4 Ω cm^2^ R_ct_ Mg(OH)_2_/sol-gel = 4.03 × 10^5^ Ω cm^2^	[116]
2010	AM60B	Composite sol-gel coating (AO/sol-gel)	Mixing together with TEOS (4.7 g), 3-metacryloxypropyl trimethoxysilane (10.4 g), ethylalcohol (15.8 g), distilled water (4.9 g), and tert-butylhydroperoxide (1.9 g).	4 μm	3.5 wt.% NaCl	i_corr_ Mg = 3 × 10^−5^ A/cm^2^ i_corr_ AO = 2 × 10^−6^ A/cm^2^ i_corr_ AO/sol-gel = 7 × 10^−9^ A/cm^2^	[117]
2009	AZ91D	Composite sol-gel coating (MAO/sol-gel)	TEOS, zirconyl chloride octahydrate (ZrOCl_2_·8H_2_O), and ethanol were mixed together.	5 μm	3.5 wt.% NaCl	i_corr_ Mg = 3.395 × 10^−5^ A/cm^2^ i_corr_ MAO = 3.921 × 10^−7^ A/cm^2^ i_corr_ MAO/sol-gel =1.577 × 10^−9^ A/cm^2^	[122]
2012	NZ30K	Composite sol-gel coating (MAO/sol-gel)	The desired amounts of TEOS, C2H5OH, NH4OH, and H2O were mixed with a molar ratio of 1:30:1:1. The calculated amount of MTES (molar ratio of MTES/TEOS = 1/2) was added dropwise into the mixed solution.	3.5–7 μm	3.5 wt.% NaCl	i_corr_ Mg = 2.2 × 10^−5^ A/cm^2^ i_corr_ MAO = 2.5 × 10^−7^ A/cm^2^ i_corr_ MAO/sol-gel =2.6 × 10^−8^ A/cm^2^ |Z|_0.01Hz_ Mg ≈ 10^3^ Ω cm^2^ |Z|_0.01Hz_ MAO ≈ 10^5^ Ω cm^2^ |Z|_0.01Hz_ MAO/sol-gel ≈ 3 × 10^6^ Ω cm^2^	[132]
2016	ZE41	Composite sol-gel coating (PEO/sol-gel)	Silane sol: mixing GPTMS and PTMS (volume ratio was 1: 1); metal organic: mixing TPOT (70 wt.% in 2-propanol) and acetylacetone in stoichiometric proportion. Both metal organic and silane sols were mixed together.	7.8–8.4 μm	3% NaCl	After 7 days of immersion, Sol-gel: about 50% of the coating was exfoliated from the surface. |Z|_0.01Hz_ MAO/sol-gel = 3 × 10^8^ Ω cm^2^	[121]
2017	AZ31	Composite sol-gel coating (MAO/sol-gel)	Polymethyltrimethoxysilane (PMTMS): (MTMS: ethanol: DI water = 3:10: 20)	13.65 μm	3.5 wt.% NaCl	i_corr_ Mg = 1.37 × 10^−5^ A/cm^2^ i_corr_ MAO/sol-gel = 2.86 × 10^−8^ A/cm^2^ R_ct_ Mg = 275.30 Ω cm^2^ R_ct_ MAO/sol-gel = 2.24 × 10^6^ Ω cm^2^	[123]
2019	AZ80	Composite sol-gel coating (PEO/sol-gel)	Ethanol: silica precursors: water: hydrochloric acid (a molar ratio) = 2:1:4:0.01. The ratio between TEOS and MTES were fixed at 30:70.	22 μm	0.1 M Na_2_SO_4_ + 0.05 M NaCl	PEO/sol-gel that is characterized by currents about two orders of magnitude lower than the untreated one.	[124]
2021	AZ31B	Composite sol-gel coating (PEO/sol-gel)	Mixing 0.5 mol TEOS, 0.5 mol GPTMS, and 0.54 mol of a colloidal SiO2 nanoparticles suspension. Ethanol containing 0.1 mol of 1-Methylimidazole (MI) were added.	3.5 μm	3.5 wt.% NaCl	i_corr_ Mg = 1.61 × 10^−5^ A/cm^2^ i_corr_ PEO = 2.64 × 10^−7^ A/cm^2^ i_corr_ PEO/sol-gel = 2.80 × 10^−8^ A/cm^2^ R_p_ Mg = 207.3 Ω cm^2^ R_p_ PEO =31432.5 Ω cm^2^ R_p_ PEO/sol-gel =31,546.8 Ω cm^2^ PEO/sol-gel includes an additional diffusive resistance (68716 Ωcm^2^) (non-faradaic resistance) between the sol-gel coating and PEO oxide layer	[125]
2021	AZ31	Composite sol-gel coating (PEO/sol-gel)	A molar fraction of 40% TEOS and 60% MTES. Diluting in isopropanol and 0.1 M of HCl acidulated H_2_O in a molar ratio of 1 mol of the mixture of precursors, 5 mol of isopropanol, and 10 mol of acidulated H_2_O. In addition, sol was doped with 0.005 wt.% Grade 4 −COOH functionalized GNPs (COOH−GNPs).	36.7 μm	Hanks’ solution (pH = 7)	After being soaked for 24 h, i_corr_ Mg = 1.5 × 10^−6^ A/cm^2^ i_corr_ PEO = 1.6 × 10^−7^ A/cm^2^ i_corr_ PEO/sol-gel = 2.50 × 10^−8^ A/cm^2^	[103]
2022	AM6	Composite sol-gel coating (PEO/sol-gel)	Mixing two different sols using controllable hydrolysis of γ-GPS and TEOS.	19.3 μm	3.5 wt.% NaCl	R_p_ PEO = 3.37 ×10^5^ Ω cm^2^ R_p_ PEO/sol-gel = 3.58 × 10^9^ Ω cm^2^	[126]
2020	AZ91D	Composite sol-gel coating (N-GQDs/sol-gel)	PMTMS: (MTMS: ethanol: DI water = 3: 10: 20)	19 μm	3.5 wt.% NaCl	R_ct_ Mg = 78.3 Ω cm^2^ R_ct_ N-GQDs /sol-gel = 1.7 × 10^4^ Ω cm^2^	[127]
2020	AZ31	Composite sol-gel coating (PEO/sol-gel/epoxy)	T50/A50: (TEOS: APTES: Water: Ethanol =2.14:2.14:2:8(Volume ratio)	-	3.5 wt.% NaCl	After being soaked for 28 day, R_coat_ PEO/epoxy ≈ 2 × 10^6^ Ω cm^2^, R_coat_ PEO/sol-gel/epoxy ≈ 1× 10^8^ Ω cm^2^	[128]
2021	AZ31	Composite sol-gel coating (PEO/sol-gel/epoxy)	T50/A50: (TEOS: APTES: Water: Ethanol =2.14:2.14:2:8(Volume ratio) 5 ppm of organic inhibitors 8-HQ, I3C, 2- MBO, and DDTC were added individually to silane solutions.	-	3.5 wt.% NaCl/0.5 wt.% NaCl	After being soaked for 28 days in 3.5 wt.% NaCl, log |Z|_0.01Hz_ Triplex ≈ 7.5, log |Z|0.01Hz Triplex-Ce-HQ ≈ 8.8; Log |Z|_0.01Hz_ Triplex = 4.88, log |Z|0.01Hz Triplex-Ce-HQ = 6.06 with artificial defects, immersed in 0.5 wt.% NaCl solution for 48 h.	[129]
2019	AZ31B	Composite sol-gel coating (sol-gel/GO/sol-gel)	The molar ratio of mixed APS/BTESPT was ½, and the volume ratio of mixed silane:deionized water:ethanol = 1:1:8.	1100 nm	3.5 wt.% NaCl	i_corr_ Mg = 2.1852e^−4^ A/cm^2^ i_corr_ sol-gel/GO/sol-gel = 1.381e^−8^ A/cm^2^	[130]

* According to EIS test, R_tot_ = R_f_ + R_ct_ values are calculated as the sum of all the faradaic resistance by using the fitted data, |Z|_0.01Hz_ is the low-frequency impedance, R_p_ is polarization resistance, and R_ct_ is related to the charge transfer resistance. i_corr_ is the corrosion current density.

## Data Availability

The data presented in this study are available on request from the corresponding author.

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
