# Peer review of "Advances in the Modification of Silane-Based Sol-Gel Coating to Improve the Corrosion Resistance of Magnesium Alloys"

_molecules, 2023, doi:10.3390/molecules28062563_

Round 1
Reviewer 1 Report
In the review article titled "Advances in modification of silane-based sol-gel coating improving the corrosion resistance of magnesium alloys" authors presented their extensive literature-supported work. The authors also emphasized the importance of coating versatility in terms of corrosion resistance in magnesium-based alloys. They suggested that sol-gel coatings provide beneficial properties in terms of corrosion inhibitor with in-depth sol-gel coatings, with analyzes of sol-gel technology and related characterization techniques revealed that they have wider and more practical applications. The handling of the study, the organization of the presentation and its main purpose were well emphasized, and it was an article that could attract the attention of the researchers. It is at a level that can be published with very small additions.
1. It would be much better if the reactions were added in schematic steps in Figure 1.
2. In Figure 2, it would be more appropriate to write the required reaction for corrosion products under the figure.
3. If information is given about the standard electrode potentials of the galvanic corrosion conditions between the sol-gel coating and the substrate mentioned in the article, it can enrich the article.
Author Response
Dear Prof./Dr.:
We would like to thank the Reviewer for his/her thoughtful comments and efforts toward improving our manuscript.
-
1. It would be much better if the reactions were added in schematic steps in Figure 1.
Thanks for your kind suggestions. We have drawn Figure 1, which is more detailed about the sol-gel reactions.
2. In Figure 2, it would be more appropriate to write the required reaction for corrosion products under the figure.
Thanks for your kind suggestions. The corresponding figures are modified.
3. If information is given about the standard electrode potentials of the galvanic corrosion conditions between the sol-gel coating and the substrate mentioned in the article, it can enrich the article.
Thanks for your kind suggestions. We agree with your suggestions. However, in our review, much literature did not provide the electrode potential information.
Reviewer 2 Report
1. There are so many Eenglish grammer mistakes through out the manuscripts.
2. The review is not properly written
3. The Basic sol-gel reaction equation 1 and 3 are wrong
Author Response
Dear Prof./Dr.:
We would like to thank the Reviewer for his/her thoughtful comments and efforts toward improving our manuscript.
1. There are so many English grammar mistakes throughout the manuscripts.
Thanks for your kind suggestions. A native speaker helped us in modifying the grammar mistakes.
2. The review is not properly written
Thanks for your kind suggestions. This review is based on the modification methods of silane-based sol-gels on the surface of Mg alloys, which are divided into four categories: bare sol-gel, nanoparticles, corrosion inhibitors, and sol-gel-based composite coatings.
3. The Basic sol-gel reaction equation 1 and 3 are wrong
Thanks for your kind suggestions. We have made changes. Please check Figure 1.